# Smart Farming Revolution: Portable and Real-Time Soil Nitrogen and Phosphorus Monitoring for Sustainable Agriculture

**DOI:** 10.3390/s23135914

**Published:** 2023-06-26

**Authors:** Harpreet Singh, Nirmalya Halder, Baldeep Singh, Jaskaran Singh, Shrey Sharma, Yosi Shacham-Diamand

**Affiliations:** 1Faculty of Engineering, Tel Aviv University, Aviv 6997801, Israel; 2TAU/TIET Food Security Centre of Excellence, Thapar Institute of Engineering and Technology, Patiala 147004, India

**Keywords:** Arduino Uno, IoT, LDR, LED, macro-nutrients

## Abstract

Precision agriculture is crucial for ensuring food security in a growing global population. Nutrients, their presence, concentration, and effectiveness, are key components in data-driven agriculture. Assessing macro and micro-nutrients, as well as factors such as water and pH, helps determine soil fertility, which is vital for supporting healthy plant growth and high crop yields. Insufficient soil nutrient assessment during continuous cropping can threaten long-term agricultural viability. Soil nutrients need to be measured and replenished after each harvest for optimal yield. However, existing soil testing procedures are expensive and time-consuming. The proposed research aims to assess soil nutrient levels, specifically nitrogen and phosphorus concentrations, to provide critical information and guidance on restoring optimal soil fertility. In this research, a novel chip-level colorimeter is fabricated to detect the N and P elements of soil onto a handheld colorimeter or spectrophotometer. Chemical reaction with soil solution generates color in the presence of nutrients, which are then quantitatively measured using sensors. The test samples are collected from various farmlands, and the results are validated with laboratory analysis of samples using spectrophotometers used in laboratories. ANOVA test has been performed in which F value > 1 in our study indicates statistically significant differences between the group means. The alternate hypothesis, which proposes the presence of significant differences between the groups, is supported by the data. The device created in this paper has crucial potential in terms of environmental and biological applications.

## 1. Introduction

Nitrogen and phosphorus are essential nutrients for plant growth, and they play a critical role in the soil-plant system. Nitrogen and phosphorus are primarily found in the soil, but their concentration varies depending on the type of soil, location, and environmental factors. The availability of these nutrients in the soil affects plant growth and productivity. Therefore, in precision agriculture, detecting nitrogen and phosphorus is crucial to determine the quality and quantity of these nutrients in the soil. The goal is to optimize crop yield while minimizing fertilizer use. The demand for food in many countries will increase 70 percent by 2050 [1]. To address this problem, traditional agricultural processes must be automated and sped up in order to fulfill the demand of society. IoT technologies have started a new era of sensor networks in agriculture, though relatively few have been put into practice. The majority of the studies state that a wireless sensor network collects data and sends them via wireless protocol to a central server [2,3]. The information is centered on environmental circumstances, which aids in monitoring. Some publications focus on employing a sensor and webcam to identify rodents or intruders in the field using an automated security system [4,5]. All of this relies on parameter monitoring after the crop has been sowed. The state of the soil and the nutrient levels present in it receive no consideration.

In most countries, farmers continue cultivating crops in the same old and outdated ways, adding fertilizers and water to the soil based on their rough estimates rather than assessing the soil’s condition [6]. Testing the quality of agricultural soil plays a significant role in increasing the yield of crops. Agricultural soil quality is a measurement of the soil’s ability to serve as a suitable growth medium for crops by supplying the essential levels of nutrients. Crops collect nutrients from the soil that they require for growth; these nutrients are divided into macro and micronutrients [7]. Micronutrients are those that are required in lower amounts, whereas macronutrients are those that are required in large amounts. The primary macronutrients are nitrogen, phosphorus, and potassium. Calcium, sulfur, and magnesium are secondary macronutrients. The fundamental purpose of soil testing is to assure efficient and effective fertilizer management, as well as to determine the optimum dose levels required to achieve maximum crop yield [8].

Several methods have been developed for the detection of nitrogen and phosphorus in soil. These methods can be broadly classified into two categories: traditional methods and modern methods. Traditional methods for the detection of nitrogen and phosphorus in soil are based on chemical reactions. These methods have been used for decades and are still widely used due to their simplicity and cost-effectiveness. However, traditional methods have some limitations, such as low sensitivity and specificity and being time-consuming [9]. The Kjeldahl method is one of the traditional methods used to determine the total nitrogen content in the soil. In this method, a sample of soil is digested with sulfuric acid and heated to release ammonia, which is then titrated with a standard acid solution [10]. This method provides a measure of the total nitrogen content in the soil. However, it cannot distinguish between different forms of nitrogen, such as nitrate, ammonium, or organic nitrogen [11]. The other traditional method is the molybdate method which is widely used for the determination of total phosphorus in the soil. In this method, the sample of soil is mixed with sulfuric acid, and then a molybdate solution is added to the sample, which forms a complex with the phosphate ions [12]. The complex is then reduced with ascorbic acid, and the intensity of the resulting blue color is measured spectrophotometrically. This method provides a measure of the total phosphorus content in the soil [13].

Several modern methods are also used for the detection of nitrogen and phosphorus in soil based on advanced technologies such as spectroscopy, chromatography, and electrochemical techniques. These methods are susceptible, specific, and require less time and labor compared to traditional methods [14]. Some of the modern methods are spectroscopic techniques such as Fourier transform infrared spectroscopy (FTIR), Raman spectroscopy, and near-infrared spectroscopy (NIR) have been developed to detect nitrogen and phosphorus in soil. These techniques are non-destructive and do not require any sample preparation. NIR spectroscopy is widely used due to its simplicity, speed, and cost-effectiveness [15]. In this method, a sample of soil is illuminated with near-infrared radiation, and the reflected light is analyzed to determine the absorption spectra. The spectra are then correlated with the concentration of nitrogen and phosphorus in the soil [16].

Other modern methods include chromatographic techniques such as high-performance liquid chromatography (HPLC) and gas chromatography (GC) for the detection of nitrogen and phosphorus in soil. These techniques are highly sensitive and specific and can distinguish between different forms of nitrogen and phosphorus. In HPLC, a sample of soil is extracted using a solvent, and the extract is then analyzed by a column containing a stationary phase and a mobile phase [17]. The mobile phase carries the analyte to the detector, where it is quantified. In GC, the sample is vaporized and passed through a column where the analyte is separated based on its affinity to the stationary phase. The separated analyte is then detected by a detector, and its concentration is determined [18].

Electrochemical tests are also used to test soil in labs. It is a two-part procedure. To begin, a soil sample is collected in a test tube; then, chemicals are added to produce various colors for identifying various nutrients in the soil. The soil sample is next subjected to a quantitative analysis using titration, which determines the exact amount of nutrient present in the soil. The present methods of soil quality testing take time; usually, the need for a few additional days is cumbersome. So, farmers avoid soil testing and add fertilizers based on their experience and estimation [19]. The current methods of soil quality testing take a long time, usually a few days, and are inconvenient. As a result, farmers avoid soil testing and instead rely on their knowledge and estimation when applying fertilizers [20].

In recent years, there has been significant development in sensor technology for monitoring the quantity of nitrogen (N) and phosphorus (P) in soil, commonly referred to as NP monitoring. These advancements have been driven by the increasing demand for precision agriculture and the need to optimize nutrient management practices in order to enhance crop productivity and minimize environmental impact. NP monitoring sensors offer real-time, non-destructive, and accurate measurements of nutrient levels in the soil. These sensors utilize various techniques such as spectroscopy, electrochemical analysis, and ion-selective electrodes to detect and quantify NP concentrations. Spectroscopic techniques, such as near-infrared (NIR) spectroscopy, measure the absorption or reflectance of light to determine nutrient levels. Electrochemical sensors utilize electrodes to measure the electrical properties of the soil solution and calculate nutrient concentrations. Ion-selective electrodes specifically target individual nutrients and provide selective measurements. The need for NP monitoring sensors arises from several key factors. Firstly, efficient nutrient management is crucial for sustainable agriculture. Nutrients play a vital role in plant growth and development, and imbalances or deficiencies can lead to reduced yields and crop quality. By accurately monitoring NP levels, farmers and agronomists can make informed decisions about fertilizer application, ensuring that crops receive the right amount of nutrients at the right time. Secondly, excessive fertilizer application can result in nutrient runoff, leading to water pollution and environmental degradation. NP sensors enable precise nutrient monitoring, allowing farmers to apply fertilizers judiciously and minimize nutrient losses. This contributes to improved environmental stewardship and compliance with regulations governing nutrient management. Furthermore, NP monitoring sensors provide a time-saving alternative to traditional soil testing methods. Traditional soil analysis involves sample collection, transportation to laboratories, and subsequent analysis, which can be time-consuming and delay decision-making processes. In contrast, the proposed NP sensor will provide on-the-spot measurements, allowing farmers to make timely adjustments to their fertilization strategies and maximize crop productivity. Overall, the development and implementation of NP monitoring sensors offer several benefits to agricultural systems, including optimized nutrient management, reduced environmental impact, improved resource efficiency, and enhanced crop productivity. As sensor technology continues to evolve, it is expected that NP monitoring will become an integral component of precision agriculture, aiding farmers in achieving sustainable and efficient nutrient management practices.

A simple and adequate method for soil nutrient analysis is necessary due to the extensive time consumption and high cost associated with the existing methods. The current approaches require sampling, transportation to laboratories, complex procedures, and the involvement of specialized personnel, all of which contribute to prolonged analysis timelines and increased costs. By developing a simple and adequate method, these drawbacks can be addressed. Such a method would offer quick and on-site analysis, reducing the time needed to obtain results. It would also be cost-effective, minimizing the reliance on laboratory facilities and skilled technicians. This would enable farmers and agricultural professionals to make timely decisions regarding fertilizer application and nutrient management, ultimately leading to improved crop productivity and resource efficiency.

The goal of this study is to create a wireless sensor system that can detect soil conditions in real time and calculate the best time to administer fertilizers to boost productivity. A chemical kit, as well as a wireless sensor network consisting of a pocket-sized device with LDR and LED, is interfaced with an Android-based application utilizing an Arduino uno and a Wi-Fi module in this study.

## 2. Materials and Methods

### 2.1. Methodology Workflow

Representative soil samples were collected from the study area using standard soil sampling techniques, ensuring random sampling across the entire site. The soil samples were processed by extracting the supernatant through a standardized procedure. The supernatant was then mixed with a predetermined chemical compound known for its affinity to nitrogen (N) and phosphorus (P) compounds. The resulting mixture was carefully transferred into a clean and transparent glass cuvette, ensuring minimal air bubbles or contaminants. The cuvette containing the solution was placed inside a high-resolution spectral analyzer. The spectral analyzer was calibrated and set to the appropriate wavelength range to measure the absorption or transmission of light by the solution. The spectral analyzer recorded the optical properties of the solution at the designated wavelengths. A series of measurements were taken to ensure reliable and accurate data acquisition. The acquired spectral data were subjected to regression analysis using a suitable mathematical model. The model was developed based on a calibration dataset with known concentrations of N and P, allowing for the determination of their respective contents in the soil samples. Based on the regression analysis, the concentrations of available nitrogen and phosphorus in the soil samples were calculated and expressed in parts per million (ppm). The calculated concentrations of available nitrogen and phosphorus in ppm were then converted to kilograms per hectare (Kg/h) using appropriate conversion factors. This conversion allowed for a more practical and meaningful representation of nutrient content relevant to agricultural practices, as shown in Figure 1.

### 2.2. Color Intensity-Based Detection

In the study, the device utilized a Light Dependent Resistor (LDR) sensor to detect color intensities. The LDR sensor is a commonly used component that changes its resistance based on the amount of light falling on it. By incorporating this sensor into the device, it was able to measure the intensity of the light transmitted through the developed soil sample solution. To elaborate further, when the cuvette with the developed soil sample solution is placed inside the device, the LDR sensor captures the light transmitted through the solution. The intensity of the transmitted light is inversely proportional to the color intensity of the developed soil solution. Darker colors absorb more light and transmit less, while lighter colors transmit more light. The LDR sensor detects these variations in light intensity and converts them into electrical signals. The device’s circuitry then processes the electrical signals produced by the LDR sensor and converts them into a numerical scale that ranges from 0 to 1024. This numerical scale represents the color intensities of the developed soil solution, with higher values indicating higher transmittance, lower nitrogen or phosphorus content, and vice versa. Therefore, the device provides a quantitative measurement of the color intensities, allowing users to assess the nitrogen and phosphorus content in the soil based on the detected values. By using the LDR sensor, the device offers a practical and relatively simple approach to detecting color intensities in developed soil samples.

### 2.3. Chemicals

In this study, it was ensured that all chemicals used for experimental purposes were of AR grade (analytical reagent grade) and conformed to reagent grade standards. AR grade chemicals are of high purity and are commonly used in analytical and research laboratories. These chemicals undergo rigorous quality control measures to meet specific standards and have a high level of consistency in their composition and properties. By using AR grade chemicals, the study aimed to minimize potential sources of variability or impurities that could affect the experimental results. This ensures that the chemicals used in the study are reliable, consistent, and suitable for the intended analytical or experimental procedures. Conforming to reagent grade standards implies that the chemicals meet the specific requirements for their designated applications. Reagent grade chemicals are commonly used in laboratory settings and are held to defined quality standards, ensuring their suitability for various analytical techniques or experimental protocols. By explicitly stating the utilization of AR grade chemicals that conform to reagent grade standards, the study emphasizes the adherence to high-quality standards, precision, and accuracy in the experimental procedures. This ensures the reliability and validity of the results obtained from the study.

Phenol, sulphuric acid, brucine, chloroform, potassium antimony tartrate, ammonium molybdate, ascorbic acid, potassium chloride and Sodium bi-carbonate, sodium nitrate, Di-sodium hydrogen phosphate were purchased from Loba Chemie. All chemicals were reagent grade.

### 2.4. Electronic Parts and Software

Arduino microcontroller, Wi-Fi Transceiver module, RGB common cathode LED, Light Dependent Resistor or photoresistor were procured from Robu.in. AutoCAD and Android Studio were used for developing the proposed device. All spectrophotometric readings were taken in Thermo multiscan GO equipped with Skanit RE 5.0 software. Statistical analysis was done by using OriginPro 2018 edition. Voltage required to power the device is 5V. The proposed device consists of various components of very low-cost price, such as a single PCB board, an LED, a resistor, an LDR sensor, wires an Arduino Uno as the main control unit, a Bluetooth module for wireless communication, and a glass cuvette. By considering the individual costs, we can calculate the total expenditure for the device. The estimated total cost for the proposed device is approximately twenty-five US dollars. The brands used for the electronic parts are mostly generic, as shown in Table 1.

### 2.5. ANOVA Method Analysis

The ANOVA method is performed here to validate the proposed device’s variance with different data collection groups. This statistical analysis allows us to determine whether significant differences exist in the means of the groups being compared. Upon analyzing the collected data, it becomes evident that the mean values of the different groups are not the same, indicating the presence of variations [21]. This observation aligns with the alternative hypothesis, which suggests significant differences between the groups.

To further evaluate the significance of these differences, we calculate the F-value, which measures the ratio of the variation between group means to the variation within each group. When the F-value is greater than 1, it suggests significant differences between the groups being compared. In our analysis, the calculated F-value for nitrogen is 1.05, and for phosphorus, it is 1.03. Both of these values are greater than one, as shown in the Table 2 and Table 3. In other words, it indicates a significant difference between the means of nitrogen and phosphorus values across the different groups or treatments being compared. Based on these findings, the null hypothesis, which assumes no significant difference between the groups, is rejected. This suggests that the proposed device significantly affects the soil’s measured nitrogen and phosphorus levels. The observed variations in the data can be attributed to the functioning of the device rather than random fluctuations.

Utilizing the ANOVA method, we have successfully validated the proposed device for measuring the amount of nitrogen and phosphorus in soil. This statistical analysis provides strong evidence to support the effectiveness and reliability of the device in accurately assessing nutrient levels in soil samples.

### 2.6. Collection and Processing of Soil

The soil was collected randomly from specified agriculture fields in a zig-zag pattern with the help of a soil auger. Auger was pushed (up to the mark), turned into the soil to the desired depth, lifted to remove the core, and placed on the clean plastic sheet. Sampling depth should be 3 to 6 inches deep for annual crops for turf, lawns or tillage depth (usually 6–10 inches). After breaking the clumps of soil, all soil samples were mixed thoroughly over the plastic sheet to prepare a composite sample, which was further tested.

### 2.7. Preparation of Standard Nitrogen and Phosphorus Solution

Di-potassium hydrogen phosphate was taken for preparation of standard phosphate solution. Sodium nitrate was taken for preparation of the standard nitrate solution. An initial stock solution of 1000 ppm was made from which further dilutions were made using the following Equation (Equation 1),
(1)$$V_{1}\cdot S_{1}=V_{2}\cdot S_{2}.$$
where,
$V_{1}$ = Amount to be taken from stock$S_{1}$ = Concentration of stock$V_{2}$ = Amount of solution of new concentration to be made$S_{2}$ = Concentration of solution to be made


The concentrations of different standard nutrient solutions were made according to specific nutrient requirement levels of different agricultural crops.

Chemical tests for color development were performed according to the modified Strickland and Parsons method [22,23] and modified Phenol-Disulphonic acid method [24,25,26] for phosphate and nitrate content respectively.

### 2.8. Design of Proposed Device

The design of an outer structure or casing of the hardware was created using AutoCAD as shown in Figure 2. It was further uploaded and printed using the 3-D printing service provided by Robu.in (Pimpri-Chinchwad, India). Circuitry was built using an Arduino microcontroller with a Wi-Fi Transceiver module. An RGB common cathode LED and a Light Dependent Resistor or photoresistor were used as input and output sensors, respectively. The entire software was developed in the C programming language on the Arduino interface. Finally, using an Android app created in Android Studio, the results were retrieved and displayed.

## 3. Results and Discussion

The device was developed by 3-D printing in three parts and later assembled into one, as shown in Figure 1. Inside of the lid and main body were painted dark matte black to minimize the reflections or refraction of light from the source LED to the photoresistor as shown in Figure 3.

After the development of the colored reaction with N and P nutrient components, the readings of color intensity were taken through a spectrophotometer as well as through the proposed device. Color intensity is directly proportional to the concentration of nutrient present. More the concentration of nutrient present, the higher the color intensity and the higher the reading value. An increasing pattern in values can be seen in both spectrophotometric and device values (Figure 4, Figure 5, Figure 6 and Figure 7). In the case of device readings, RGB LED light source was equipped according to the specified color code for the highest absorbance. Both solutions gave the highest absorbance of light at a different wavelength of a light source.

Phosphate solution after reaction gave blue color (Figure 4 and Figure 5). That is why in the case of spectrophotometric reading of phosphate solution, 880 nm was used, which leads to the highest absorbance. For readings of phosphate solution through the device, the LED color of the red band was found to be of imparting the highest absorption. The corresponding Hex Code and Decimal Code has been shown in Table 4.

Due to the color saturation of the sample solution, the phosphorus regression equation was divided into two parts, i.e., one for initial values from 0.0 to 2.5 and the second for 2.5 to 3.0. This was fitted in the device coding, which will run in the backend for unknown sample estimation.

In our research, we analyzed the behavior of our device by comparing it to a spectrophotometer, as depicted in Figure 4 and Figure 5. It was important to ensure that the device provided accurate readings, so we divided the graph in Figure 5 into two regions: region 1 (0–200 ppm) and region 2 (200–800 ppm). This division was based on our observations during the testing phase, which revealed that the device exhibited different behavior for concentrations beyond 200 ppm. The discrepancy in behavior can be attributed to the way the samples were prepared and the sensors used in the device. This observation is supported by the R${}^{2}$ values obtained from the graphs (Equations (Equation 4) and (Equation 5)). For the phosphorus trend within the range of 0–200 ppm, the data followed a cubic equation with an R${}^{2}$ value of 0.9969 (Equation (Equation 4)). However, beyond 200 ppm, the trend became linear with an R${}^{2}$ value of 0.9998 (Equation (Equation 5)). Therefore, depending on the concentration, either of these equations can be used to determine the device readings. Equation (Equation 2) is employed for concentrations below 200 ppm, while Equation (Equation 3) is applied for concentrations beyond that range. These equations provide readings that are comparable to real spectrometric readings.

In contrast to the phosphorus readings, the nitrate readings exhibited a different trend. Throughout the entire concentration range of 0–1000 ppm, our device generated trends that closely resembled the spectrometric readings. Figure 6 illustrates the spectrometric graph of Nitrate across various concentrations. Equation (Equation 6) represents the best-fit line for this graph, with an R${}^{2}$ value of 0.993. Figure 7 showcases the spectrometric graph generated by our device, and Equation (Equation 7) represents its best-fit equation, which yielded an R${}^{2}$ value of 0.928.

Regression analysis of the spectrophotometric phosphate curve generated the following Equations (Equation 2) and (Equation 3) and Regression analysis of the device phosphate curve generated the following Equations (Equation 4) and (Equation 5).

Regression equation of Phosphate—spectrophotometer readings (0–100 ppm),
(2)$$y=0.08462+0.01927x-3.78714E^{-6}x^{2}-1.82385E^{-7}x^{3}$$
R^2^ = 0.9969

Regression equation of Phosphate—spectrophotometer readings (200–800 ppm),
(3)$$y=2.4053+4.4925E^{-4}x$$
R^2^ = 0.9998

Regression equation of Phosphate—device readings (0–100 ppm),
(4)$$y=169.14007+7.3033x-0.03486x^{2}$$
R^2^ = 0.9983

Regression equation of Phosphate—device readings (200–800 ppm),
(5)y=598.28571+0.04571x
R^2^ = 0.8635

Similarly, in the case of nitrate solution, after reaction, it gave a yellow color (Figure 6), and 440 nm was used for nitrate readings through a spectrophotometer. For readings of nitrate solution through the device, the LED color of the blue band was found to impart the highest absorption.

Regression analysis of the spectrophotometric nitrate curve generated the following linear Equation (Equation 6), and Regression analysis of the device nitrate curve generated the following linear Equation (Equation 7).

Regression equation of Nitrate—spectrophotometer readings are:(6)$$y=0.80477+0.00133x-1.82988E^{-7}x^{2}$$
R^2^ = 0.993

Regression equation of Nitrate—device readings are:(7)$$y=623.8577+0.16526x-7.03425E^{-5}x^{2}$$
R^2^ = 0.928

Finally, all the values were fitted in the backend code of the device and made ready for unknown sample analysis. Unknown samples of soil were extracted by following standard chemical procedures. After keeping the soil solution overnight for extraction, a representative reaction volume was taken and tested for colored reaction. After color development, the solution was taken in a cuvette, and readings were taken in both spectrophotometer and device.

A known range of standard solutions (10 ppm to 1000 ppm) of phosphorus and nitrogen (Nitrate) in triplicate was taken for training the proposed device. The above-mentioned steps were performed repeatedly to reach accuracy and precision. The same known samples were also tested in a standard spectrophotometer, followed by a correlation established with both data. The standard spectrophotometric method was used to validate the data from the proposed device. Multiple known samples of different concentrations of nitrogen and phosphorus were used to establish the high-throughput nature of the device instead of unknown samples. Finally, the proposed device has been validated with unknown samples collected from the agricultural field. Various soil samples were collected from the agricultural field around different villages in Punjab, India. For test analysis, few samples tags marked as SP-1, SP-2, SP-3, and SP-4 which were collected from different GPS locations having co-ordinates of [30${}^{\circ }$20${}^{\prime }$51.8${}^{\prime \prime }$ N, 76${}^{\circ }$20${}^{\prime }$30.7${}^{\prime \prime }$ E], [30${}^{\circ }$28${}^{\prime }$08.8${}^{\prime \prime }$ N, 76${}^{\circ }$24${}^{\prime }$20.9${}^{\prime \prime }$ E], [30${}^{\circ }$25${}^{\prime }$29.3${}^{\prime \prime }$ N, 76${}^{\circ }$31${}^{\prime }$53.1${}^{\prime \prime }$ E] and [30${}^{\circ }$12${}^{\prime }$19.9${}^{\prime \prime }$ N, 76${}^{\circ }$28${}^{\prime }$52.3${}^{\prime \prime }$ E], respectively, as shown in the study area map in Figure 8.

Values from both the spectrophotometer and the device were found to be similar as shown in Table 5. Thus, farmers can easily use this low-cost proposed device on agriculture fields for real-time nutrient analysis of N and P components.

## 4. Conclusions and Future Scope

### 4.1. Conclusions

The Internet of Things is widely used to connect devices and collect information. The system is designed for real-time detection of soil parameters in farms. After collecting and analyzing the data, the algorithm is designed to provide accuracy. All the results are calculated and validated by taking several readings from the agricultural field. This project can undergo further research to improve the functionality of the device and its application areas. The optimum soil nutrients will increase the yield of the crops, which will increase the overall per capita income of farmers. The farmers will eliminate the extra addition of fertilizers into the soil using Farmtech applications that will save their capital cost as well.

The Internet of Things (IoT) has gained widespread adoption for use in connecting devices and facilitating data collection. In the context of agriculture, this technology has been leveraged to develop a system specifically designed for real-time detection of soil parameters on farms. By continuously collecting and analyzing data, the system employs an algorithm to ensure accuracy in the measurement of soil characteristics. To validate the reliability of the results, multiple readings are taken from various locations within the agricultural field. Furthermore, this project holds the potential for further research and development to enhance the functionality of the device and explore new application areas. The significance of this technology lies in its potential to optimize soil nutrients. By accurately monitoring and adjusting nutrient levels in real time, farmers can substantially improve crop yields. This increase in productivity directly contributes to enhancing the overall per capita income of farmers, leading to improved livelihoods and economic prosperity. Moreover, the implementation of the Farmtech application can help farmers save costs by eliminating the excessive addition of fertilizers into the soil. By precisely determining the required nutrient quantities, farmers can avoid wasteful practices, thus minimizing their capital expenditure on unnecessary fertilizers. In summary, the utilization of IoT in soil parameter detection offers promising benefits to the agricultural sector. By enhancing productivity, saving costs, and providing real-time insights, this technology has the potential to revolutionize farming practices, bolster farmers’ income, and contribute to sustainable agriculture.

### 4.2. Future Scope

Future research in IoT-based NPK monitoring devices can focus on several key areas. First, sensor development should aim to improve accuracy, durability, and sensitivity in measuring NPK levels, exploring nanomaterial-based or optical sensors. Data fusion and analysis techniques should be investigated to provide comprehensive NPK measurements, utilizing advanced algorithms and machine learning models to extract insights for nutrient management. Efficient and low-power wireless communication protocols need to be explored to ensure seamless connectivity and energy efficiency. Energy harvesting methods, such as solar or kinetic energy, should be researched for self-powered operation. Integration with autonomous systems, such as drones or robots, can enable real-time monitoring and intelligent decision-making for optimized fertilization. Security and privacy concerns must be addressed through secure communication protocols and data encryption. Cost optimization strategies, including scalable manufacturing techniques and low-cost components, should be pursued to make these devices more affordable. Lastly, integration with existing agricultural systems will facilitate data exchange and provide actionable insights for resource utilization and crop growth optimization, ultimately enhancing the accuracy, efficiency, and usability of IoT NPK monitoring devices to improve sustainable agricultural practices.

## Figures and Tables

**Figure 1 sensors-23-05914-f001:**
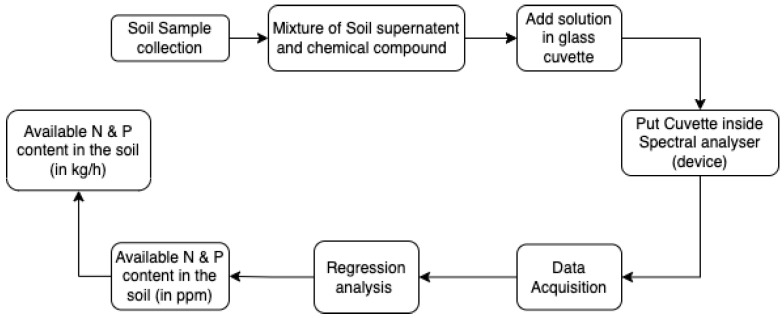
Flowchart of methodology.

**Figure 2 sensors-23-05914-f002:**
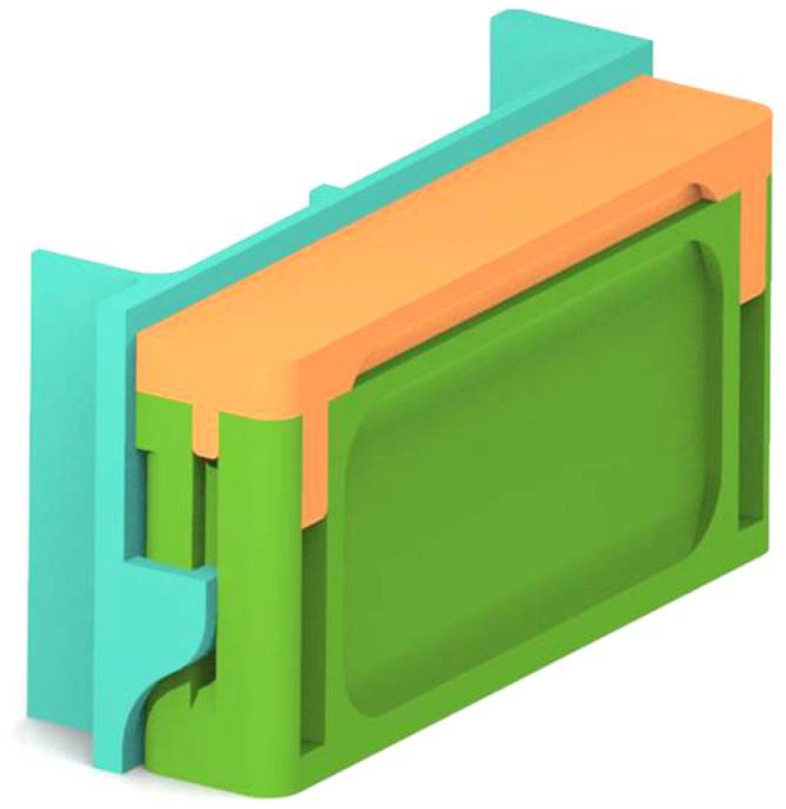
3-D design of the proposed IoT device.

**Figure 3 sensors-23-05914-f003:**
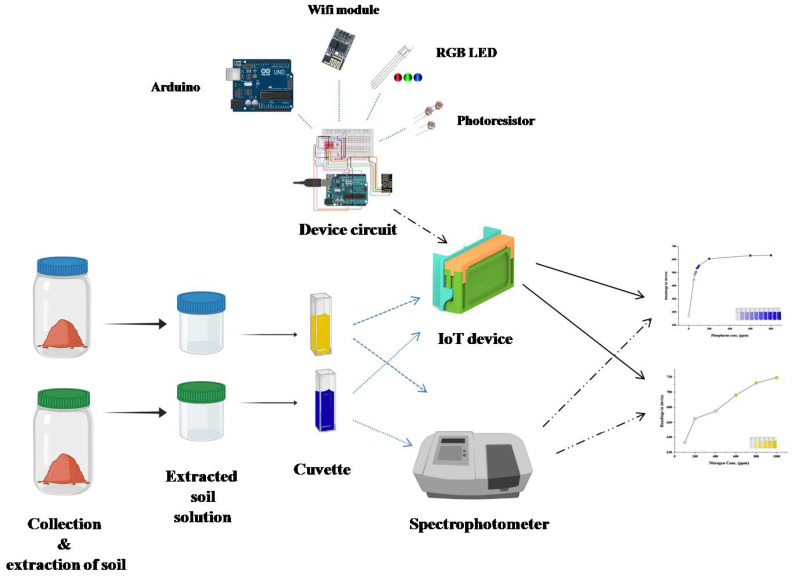
Graphical representation of proposed research.

**Figure 4 sensors-23-05914-f004:**
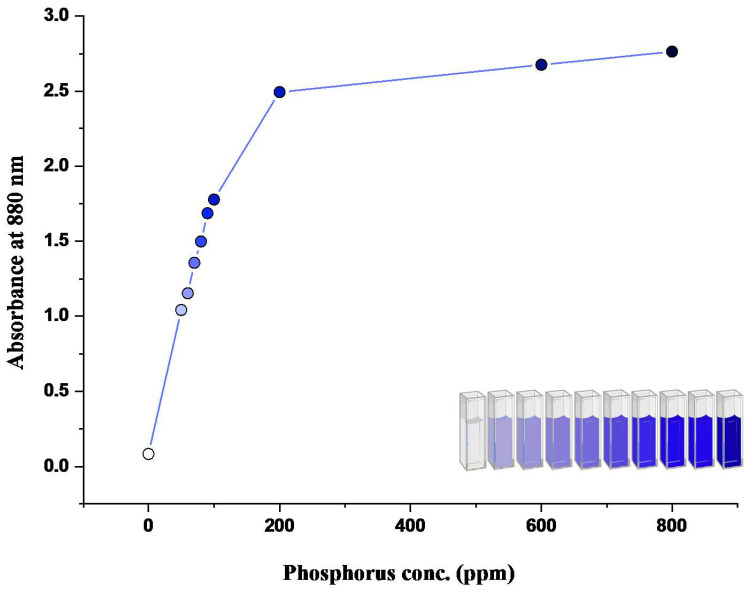
Spectrophotometric graph of phosphate.

**Figure 5 sensors-23-05914-f005:**
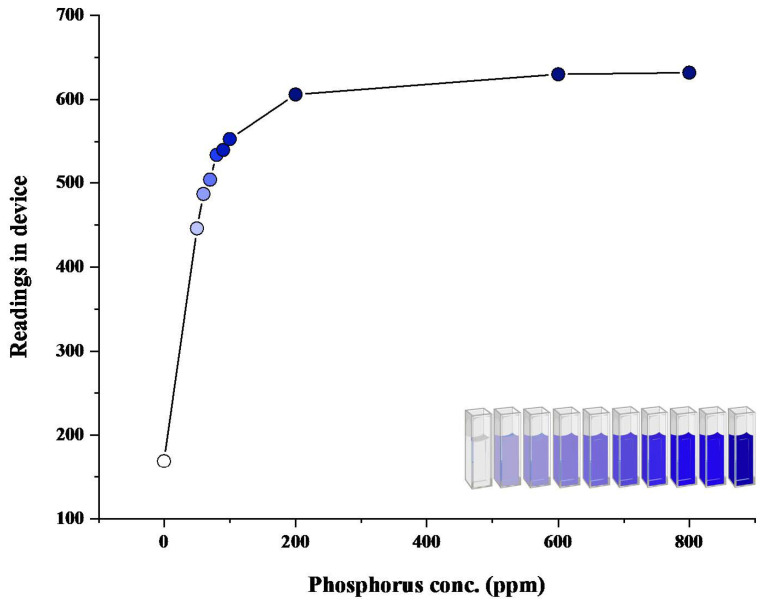
Device Generated graph of Phosphate.

**Figure 6 sensors-23-05914-f006:**
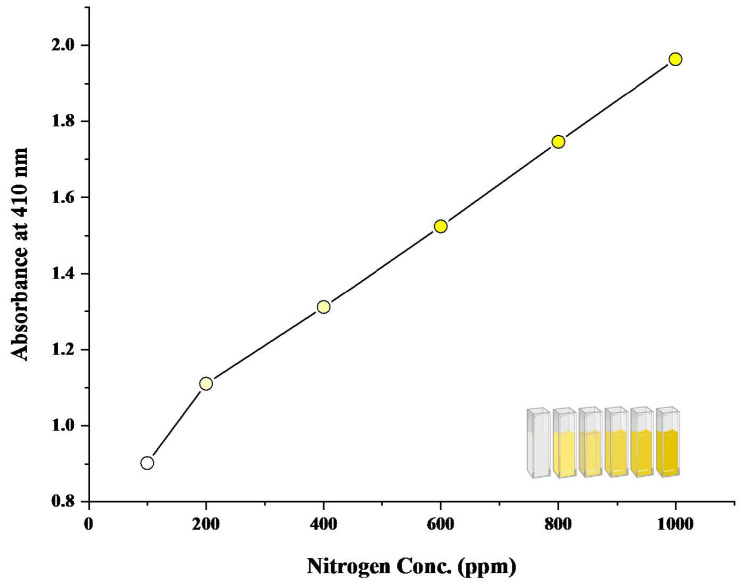
Spectrophotometric graph of Nitrate (0–1000 ppm).

**Figure 7 sensors-23-05914-f007:**
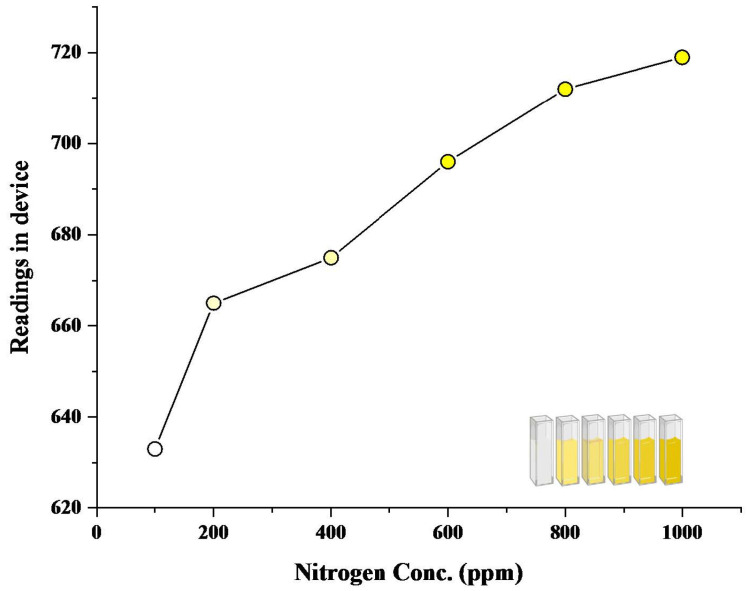
Device Generated graph of Nitrate (0–1000 ppm).

**Figure 8 sensors-23-05914-f008:**
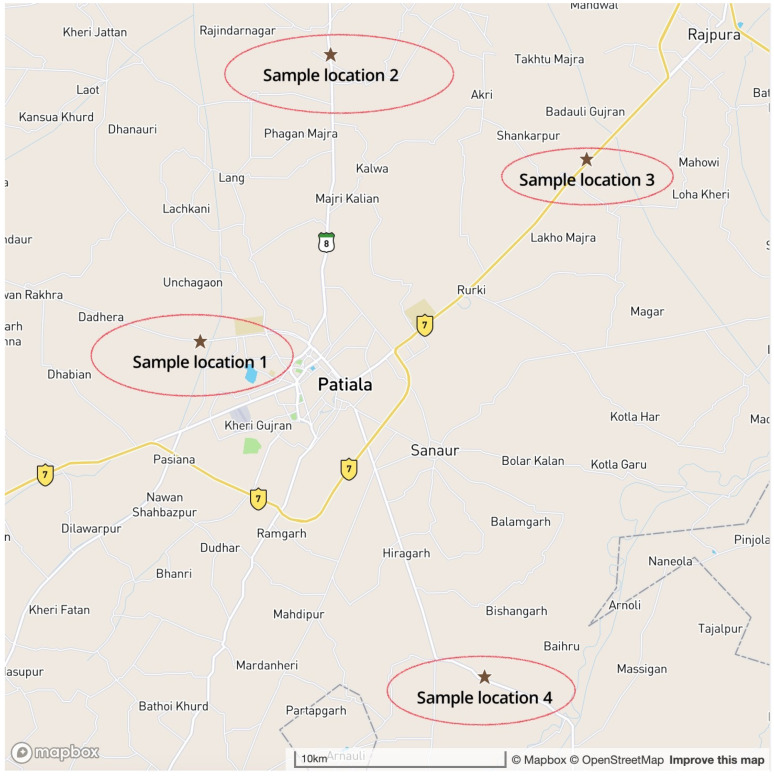
Study area map of collected soil sample.

**Table 1 sensors-23-05914-t001:** Brand of electronic parts.

Item	Brand
Wires	Genric
PCB board	Genric
LED and resistors	Genric
LDR Sensor	Genric
Arduino Uno	Arduino
Bluetooth Module	Genric
Glass Cuvette	Labrat

**Table 2 sensors-23-05914-t002:** ANOVA analysis for Nitrogen.

Nitrogen	Group A	Group B	Group C	Overall Mean	132.76
	103.88	109.88	120.88	SSB	854.78
	95.14	102.14	117.14	SSW	6075.71
	145.88	149.88	154.88	df1	2
	129.23	134.23	140.23	df2	15
	138.88	145.88	155.88	MSB	427.39
	135.23	150.23	160.23	MSW	405.04
Mean	124.70	132.04	141.54	F-value	1.05

SSB = Sum of squares between the groups. SSW = Sum of squares within the groups. MSB = Mean of the total of squares between groupings. MSW = Mean total of squares within groupings. df1 = Degree of freedom between groups. df2 = Degree of freedom Error.

**Table 3 sensors-23-05914-t003:** ANOVA analysis for Phosphorus.

Phosphorus	Group A	Group B	Group C	Overall Mean	75.14
	85.13	95.5	105.14	SSB	922.75
	45.8	50.8	57.5	SSW	6687.32
	36.1	46.1	57.3	df1	2
	72.09	80.09	95.34	df2	15
	75.1	83.1	89.15	MSB	461.38
	86.5	90.5	101.2	MSW	445.82
Mean	66.789	74.35	84.27	F-value	1.03

SSB = Sum of squares between the groups. SSW = Sum of squares within the groups. MSB = Mean of the total of squares between groupings. MSW = Mean total of squares within groupings. df1 = Degree of freedom between groups. df2 = Degree of freedom Error.

**Table 4 sensors-23-05914-t004:** Standard Red and Blue color table.

Color	Name	Hex Code (RRGGBB)	Decimal Code (R,G,B)
	Red	#FF0000	255, 0, 0
	Blue	#0000FF	0, 0, 255

**Table 5 sensors-23-05914-t005:** Comparison of available nutrients in unknown soil samples.

Sample No.	Spectro Reading		Device Reading	
Sample Tags (SP)	N	P	N	P
SP-1	115.45	1.78	642	551
SP-2	102.11	1.03	640	451
SP-3	135.78	0.93	645	427
SP-4	170.57	1.51	650	530
	Spectrophotometer Reading (calibrated in ppm) & kg/h		Proposed NP Reading (calibrated in ppm)	
Nitrogen	N (ppm)	N (kg/h)	N (ppm)	N (kg/h)
SP-1	110.88	249.5	100.4	259.8
SP-2	103.14	232.1	51.01	229.7
SP-3	149.88	337.2	44.9	305.5
SP-4	134.23	302.1	79.8	383.8
Phosphorus	P (ppm)	P (kg/h)	P (ppm)	P (kg/h)
SP-1	99.13	223.1	100.4	225.9
SP-2	50.8	114.3	51.01	114.8
SP-3	45.1	101.5	44.9	101.1
SP-4	80.09	180.3	79.8	179.6

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
