# Peer review of "Smart Farming Revolution: Portable and Real-Time Soil Nitrogen and Phosphorus Monitoring for Sustainable Agriculture"

_sensors, 2023, doi:10.3390/s23135914_

Round 1

Reviewer 1 Report

​1. The introduction requires structuring. Please establish the rationale for a simple and adequate method for soil nutrient ​analysis.    2. In the abstract, the authors should also provide the empirical data and discuss it briefly, highlighting the efficacy of the test device.    3. The authors should also provide the shortcomings of the traditional and modern methods discussed in the introduction section.   4. In order to develop a high-throughput widely acceptable monitoring system, it is necessary to test the device with multiple soil samples. Thus authors should provide details about- Number of soil samples tested Types of different soil samples tested From what depth soil samples were collected Geographic location (GPS coordinates) from where soil samples were collected.   5. A comparison of the device efficiency against the spectrophotometric method is not performed properly. In Table 2 the readers can not relate to the comparison between Spectro reading and Device reading, thus authors should only focus on Table 3 and remove table 2.    6. Most importantly, the authors should show the statistical significance of the data provided in Table 3 by showing the standard deviation and ANOVA values.

Moderate editing of English language required

Reviewer 2 Report

  1. The paper titled “Portable and real-time soil N & P monitoring system for smart agriculture described the sensors creation that has crucial potential in terms of environmental and biological applications. . The manuscript has serious flaws, lacks solids scientific model & novelty, needs extensive improvement in write up large before consideration for this journal
  2. Title is not attractive and do not use any abbreviations in the title
  3. Abstract is not written good as it reflects the review article abstract and need to change as background information is more than half of the abstract which need to be changed as need to add results and practical application of the research study
  4. What is the use of headings like “Traditional Methods” in introduction and no need to split information’s in headings rather just summarize the methods in paragraphs and then add in the introduction
  5. The major aspects like sensors technology development and application needs is lacking and need to be explained in more detail in introduction
  6. In Materials and method “All chemicals were reagent grade” need better elaborations and details
  7. Methodology and model development is not clear and needs elaboration especially development of sensors
  8. Statistical information/analysis is missing. Justify ?
  9. Do not use any abbreviations in headings “2.4. Preparation of standard N & P solution” and “2.5. Designing of IoT device”
  10. Do not use abbreviations in table headings “Table 1. Standard RGB color table” and follow for all
  11. What is the purpose of “SP-1, SP-2, SP-3 and SP-4. Justify ?
  12. What is the justification of unknown soils “Table 2. Comparison of readings of unknown soil”
  13. In conclusion major focus should be on findings with practical application as well as make it concise and it also do not reflect any practical aspects of the developed technology
  14.  
  1. Grammatical mistakes observed on several places so there is need to go through the paper for language and grammatical mistakes

Reviewer 3 Report

Comments to the authors:

The authors developed and presented a portable and real time N & P monitoring system for smart agriculture. Overall, the manuscript is methodologically sound with promising results. However, prior to further consideration, some comments to be addressed:

(i) Abstract: Include a line or two to highlight the novelty of your research.

(i) Introduction: Highlight the research gaps. Research objectives have been written but still the gap is not too clear. Also you may split line 49 to a new sub-section: Agricultural monitoring methods

Then in a separate paragraph, highlight the research objectives and research gaps.

(ii) 2.1 Chemicals: Highlight the concentration for each chemical.

(ii) Highlight the brand of the electronic parts. Different brand might lead to different results.  Provide a table.

(iii) Provide a flow chart in the methodology. Include some introduction about the system. Move it from the results and discussion to methodology. The methodology is not well understood in this study.

(iv) How did the propose device detect the colour intensities? Not well elaborated in the study.  Please provide more discussion.

(v) Highlight the voltage required for the system.

(vi) line 180-208. very confusing. Please rewrite. Give a better structure about the content.

(vii) Provide some discussion about the cost analysis.

(viii) Honestly, location in 209-212 is not really important as you are not planning to give much discussion. Remove it or add a study area map.

(ix) You can incorporate following literature to enrich the introduction: For IoT: Toward industrial revolution 4.0: Development, validation, and application of 3D-printed IoT-based water quality monitoring system.

(x) Conclusion: Conclusion section seems to be a repetition of the results section. Huge modifications are required. Please provide insights into this study and what can be further done in the future.

Please proofread the manuscript. Not scientifically sound.

Round 2

Reviewer 1 Report

The authors have thoroughly addressed all comments and the manuscript can be accepted in its present form

Author Response

Thanks for the satisfactory report.

Reviewer 2 Report

The suggested changes have been incorported so paper can be considered for publications 

The language is acceptable for scientific community

Author Response

Thanks for the satisfactory report.

Reviewer 3 Report

Comments to authors:

The authors have substantially addressed the comments which raised during the first review. The manuscript reads better but still there are some minor comments to be address:

(i) For statistical analysis, perhaps the description should be provided in the methodology section instead of the result section. Line 355-372. Please refer to : (i) Comparison among different ASEAN water quality indices for the assessment of the spatial variation of surface water quality in the Selangor river basin, Malaysia and (ii) Application of artificial intelligence methods for monsoonal river classification in Selangor river basin, Malaysia to observe how the description should be done.

(ii) Figure 8: The figure needs to have the scale bar. Also, please resize and just focus on the 4 sampling points. Only the center of the figure is important. Need not to show other areas.

Grammar: Still some minor error detected and some sentences are not scientifically sound.
